# OViTAD: Optimized Vision Transformer to Predict Various Stages of Alzheimer’s Disease Using Resting-State fMRI and Structural MRI Data

**DOI:** 10.3390/brainsci13020260

**Published:** 2023-02-03

**Authors:** Saman Sarraf, Arman Sarraf, Danielle D. DeSouza, John A. E. Anderson, Milton Kabia

**Affiliations:** 1Institute of Electrical and Electronics Engineers, Piscataway, NJ 08854, USA; 2School of Technology, Northcentral University, San Diego, CA 92123, USA; 3Department of Electrical and Software Engineering, University of Calgary, Calgary, AB T2N 1N4, Canada; 4Department of Neurology and Neurological Sciences, Stanford University, Stanford, CA 94305, USA; 5Departments of Cognitive Science and Psychology, Carleton University, Ottawa, ON K1S 5B6, Canada

**Keywords:** Alzheimer’s disease, MCI, vision transformer, rs-fMRI, MRI

## Abstract

Advances in applied machine learning techniques for neuroimaging have encouraged scientists to implement models to diagnose brain disorders such as Alzheimer’s disease at early stages. Predicting the exact stage of Alzheimer’s disease is challenging; however, complex deep learning techniques can precisely manage this. While successful, these complex architectures are difficult to interrogate and computationally expensive. Therefore, using novel, simpler architectures with more efficient pattern extraction capabilities, such as transformers, is of interest to neuroscientists. This study introduced an optimized vision transformer architecture to predict the group membership by separating healthy adults, mild cognitive impairment, and Alzheimer’s brains within the same age group (>75 years) using resting-state functional (rs-fMRI) and structural magnetic resonance imaging (sMRI) data aggressively preprocessed by our pipeline. Our optimized architecture, known as OViTAD is currently the sole vision transformer-based end-to-end pipeline and outperformed the existing transformer models and most state-of-the-art solutions. Our model achieved F1-scores of 97%±0.0 and 99.55%±0.39 from the testing sets for the rs-fMRI and sMRI modalities in the triple-class prediction experiments. Furthermore, our model reached these performances using 30% fewer parameters than a vanilla transformer. Furthermore, the model was robust and repeatable, producing similar estimates across three runs with random data splits (we reported the averaged evaluation metrics). Finally, to challenge the model, we observed how it handled increasing noise levels by inserting varying numbers of healthy brains into the two dementia groups. Our findings suggest that optimized vision transformers are a promising and exciting new approach for neuroimaging applications, especially for Alzheimer’s disease prediction.

## 1. Introduction

An early diagnosis of Alzheimer’s disease (AD) delays the onset of dementia consequences for this life-threatening brain disorder and reduces the mortality rate and billion dollars cost of caring for AD patients [1,2,3,4]. The damages that Alzheimer’s disease inflict are widespread, mostly targeting memory. Over time, shrinkage of the brain, the atrophy of the posterior cortical brain tissue degradation in the right temporal, parietal, and left frontal lobes and ventricular expansion interfere with patients’ language, and memory abilities [5,6,7]. Researchers consider a transition phase known as mild cognitive impairment (MCI) from normal aging to acute AD, which often takes two to six years. As a result, patients lack focus, exhibit poor decision-making and judgment, experience time and location confusion, and suffer the onset of memory loss [8,9,10].

Among various biomarkers examinations such as blood and clinical tests, neuroimaging has remained the primary approach for medical practitioners to attempt an early prediction of Alzheimer’s disease [11,12,13,14]. However, neurologists conduct various neuroimaging tests to diagnose Alzheimer’s disease, since the impact of normal aging and early-stage Alzheimer’s are barely distinguishable in neuroimaging [15].

Today, artificial intelligence (AI) in neuroimaging is considered an emerging technology where neuroscientists employ and adapt novel and advanced algorithms to analyze medical imaging data [16,17,18]. Over the past decade, deep learning techniques have enabled medical imaging scientists to predict various stages of Alzheimer’s disease [19,20]. Using robust computational resources such as cloud computing, the scientists could implement end-to-end prediction pipelines to preprocess medical imaging data, build complex deep learning models, and post-process results to assist medical doctors in distinguishing early-stage MCI brains from highly correlated normal aging images [21,22,23,24].

Convolutional neural networks (CNNs) inspired by the human visual system form the core image classification component of pipelines. CNN-based classifiers consist of sophisticated feature extractors that retrieve hierarchical patterns from brain images and produce highly accurate predictions [25,26,27,28,29,30]. Although CNN models often require a light preprocessing pipeline, and the models aim to lessen the impact of noise implicitly, many studies have shown that a comprehensive preprocessing pipeline, to prepare neuroimaging data significantly improves prediction performance [31,32,33].

Advances in CNNs architectures and hybrid CNNs with other architectures, such as recurrent neural networks (RNNs), have significantly improved the performance and multi-stage AD prediction [34,35,36]. The central pillar of CNN-based pipelines is the convolutional layer, considered an invariant operator in signal and image processing. The convolutional layer reduces the sensitivity of the image classification pipeline to morphological variations such as shift and rotation [37,38,39].

Also, multi-dimension filters in CNN models and various combinations of feature map concatenation enhance such models’ invariant characteristics [40,41]. However, the high complexity of models with hundreds of millions of trainable parameters, requiring high computations with an enormous amount of data, is considered a disadvantage of such methods [42,43,44]. Moreover, CNN models incorporate contextual information into training without considering positional embedding offered by transformer block [45,46].

In this study, we explore a novel method to bridge the gap of position-based context extraction through an optimized vision transformer. The literature review shows that using the vision transformer in predicting Alzheimer’s disease is in a very early stage, and this study opens a new avenue to employ vision transformers in this domain. We implement two separate end-to-end pipelines to predict the triple class of Alzheimer’s disease stages where pre- and postprocessing modules play crucial roles in improving prediction performance. Also, we analyze the impact of merging MCI data with healthy control and Alzheimer’s brains to analyze modeling performance in binary classification tasks.

We repeat each model three times using random data splits and assess our pipelines using standard evaluation metrics by averaging across repetitions to ensure the robustness and reproducibility of our models. Finally, we visualize the attention and features maps to demonstrate the global impact of attention mechanisms employed in the architecture.

## 2. Related Work

Machine learning applications in predicting various stages of Alzheimer’s disease have been of interest to numerous researchers, who began employing classical techniques such as support vector machines [47,48]. Researchers extracted features from Alzheimer’s imaging data using autoencoders and classical techniques to classify AD and MCI brains. This approach introduced more advanced feature extractors compared to the classical methods, which improved the performance of AD prediction [49,50]. The next generation of predictive models included many CNN architectures to classify mainly AD and HC brains. The successful binary classification motivated imaging and neuroscientists to employ sophisticated techniques to address 3-class prediction tasks of HC vs. MCI vs. AD [51,52,53,54].

Besides 2D CNN architectures, 3D convolutional layers enabled scientists to incorporate the volumetric data into the training process. Such approaches produced promising predictions using structural MRI data [55,56,57]. The 3D models used the signal intensity at the voxel level and applied the convolution operator to 3D filters and previous-layer feature maps. Although 3D models became popular due to producing high accuracy rates, many scientists challenged these techniques, conducting experiments in which 2D models outperformed 3D models [54,58,59,60].

Some research groups considered using functional MRI 4D data to predict various stages of AD, where they composed the brain images into 2D samples along with depth and time axes. The data decomposition method produced a significant amount of data for training and resulted in a nearly perfect binary classification performance, outperforming most of the models built by structural data. The major challenge in using 4D fMRI data was to establish a preprocessing pipeline to prepare the data for model development [61,62,63,64].

Recurrent neural networks (RNNs) and their subsequent architectures, such as long- and short-term memory LSTM models, capture features from a sequence of data that are useful to extract temporal relationships encoded in Alzheimer’s imaging data [65,66]. A special use of LSTM models occurs in longitudinal analysis for Alzheimer’s disease prediction. In this approach, researchers extract spatial maps from imaging data using various feature extractors, such as multi-layer perceptron (MLP), and train bi-directional LSTMs to address the AD classification problems. This two-step prediction allows neuroscientists to explore the patterns in longitudinal imaging, which are suppressed in cross-sectional methods [67,68,69]. However, the extra step of explicit feature extraction and the complex impact of sensitive longitudinal analysis remains the major challenge of using such methods [70,71].

The next category of machine learning methods used for Alzheimer’s disease prediction is hybrid modeling, where CNNs and RNNs models extract hierarchical and temporal features in a cascade architecture. The CNN component of such networks is considered the central feature extractor, and the RNN-LSTM component extracts position-related features and forms the core of the model [35,71,72].

Multimodal imaging in the same category provides complementary information from each modality, such as fMRI, structural MRI, and PET, that often transfer the predicted labels to a postprocessing or ensemble model. Since the nature of each modality is different, using combined data to build a unique model for AD prediction produces poor performance because the model hardly converges [73,74,75,76]. Some researchers considered a hybrid approach, using clinical and imaging data to develop separate models that followed a predictive model. Such a technique offers strong decision-making, since the misprediction by imaging models is compensated for by clinical data [77,78].

Transformers with the various implementations of attention mechanisms stemming from natural language processing (NLP) domains have been of interest to scientists regarding whether such technology is adaptable for Alzheimer’s disease prediction [79,80]. For example, a deep neural network with transformer blocks was the core of an Alzheimer’s study to assess risks using targeted speech [81].

Transformers’ temporal or sequential feature extraction capability allowed researchers to develop end-to-end solutions to predict Alzheimer’s through a longitudinal model known as TrasforMesh using structural data [82]. Also, a universal brain encoder based on a transformer with attention mechanisms offered model explainability to analyze 3D MRI data [82]. The transformer technology has motivated scientists to implement predictive models using 3D data in non-Alzheimer’s studies, such as defect assessment of knee cartilage [83]. To date, our proposed method of using an optimized vision transformer (OViTAD) to predict various stages of Alzheimer’s is considered the first initiative in adopting this technology.

## 3. Materials and Methods

### 3.1. Datasets

We used two sets of Alzheimer’s Disease Neuroimaging Initiative (ADNI) database (http://adni.loni.usc.edu/, 15 July 2021), including fMRI and structural MRI imaging data. We recruited older adults (age group > 75) for both imaging modalities in this study with the aim of suppressing the effect of aging on modeling. Using only older adults in this study enabled us to ensure our models predict the Alzheimer’s stages not aging effect. We ensured ground truth quality; we cross-checked the participants’ proposed labels by ADNI with their mini-mental state examination (MMSE) scores. The fMRI dataset contained 275 participants scanned for resting-state fMRI (rs-fMRI) studies; we found 52 Alzheimer’s (AD), 92 healthy control (HC), and 131 MCI brains in our fMRI dataset. The structural MRI dataset included 1076 participants, where we found 211 AD, 91 HC, and 744 MCI brains. Table 1 shows the participants’ demographic details for both modalities categorized into three groups: gender, age, and MMSE scores.

### 3.2. Image Acquisition Protocol

ADNI provided a standard protocol to scientists to acquire imaging data using three Tesla scanners, including General Electric (GE) Healthcare, Philips Medical Systems, and Siemens Medical Solutions machines [84]. We ensured that the two datasets utilized in this study were collected using the same scanning parameters. The protocol stated that the functional scans were performed using an echo-planar imaging (EPI) sequence (150 volumes, repetition time (TR) = 2 second (s), echo to time (TE) = 30 milliseconds (ms), flip angle (FA) = 70 degrees, filed-of-view (FOV) = 20 centimeters (cm)) that produced 64 × 64 matrices with 30 axial slices of 5 millimeters (mm) thickness without a gap. The structural MRI data acquisition employed a 3-dimensional (3D) magnetization prepared rapid acquisition gradient echo sequence known as MPRAGE (TR = 2 s, TE = 2.63 ms, FOV = 25.6 cm) that produced 256 × 256 matrices with 160 slices of 1mm thickness.

### 3.3. Data Preprocessing

#### 3.3.1. rs-fMRI

We considered an extensive 7-step pipeline to preprocess the rs-fMRI data to preprocess our data from scratch, as the research indicated that enhanced preprocessing rs-fMRI data improved the performance of modeling [85,86]. First, we converted the raw rs-fMRI data, downloaded from ADNI in digital imaging and communications in medicine (DICOM) format, to neuroimaging informatics technology initiative (NIfTI/NII) format using an open-source tool known as the dcm2niix software [87]. We removed skull and neck voxels considered non-brain regions from the structural T1-weighted imaging data corresponding to each fMRI time course using FSL-BET software [88]. Third, using FSL-MCFLIRT [89], we corrected the rs-fMRI data for motion artifact caused by low-frequency drifts, which could negatively impact the time course decomposition. Finally, we applied a standard slice timing correction (STC) method known as Hanning-Windowed Sinc Interpolation (HWSI) to each voxel’s time series. According to the ADNI data acquisition protocol, the brain slices were acquired halfway through the relevant volume’s TR; therefore, we shifted each time series by a proper fraction relative to the middle point of TR period. We spatially smoothed the rs-fMRI time series using a Gaussian kernel with 5 mm full width half maximum (FWHM). Next, we employed a temporal high-pass filter with a cut-off frequency of 0.01 HZ (Sigma = 90 s) to remove low-frequency noise. We registered the fMRI brains to the corresponding high-resolution structural T1-weighted scans using an affine linear transformation with seven degrees of freedom (7 DOF). Subsequently, we aligned the registered brains to the Montreal Neurological Institute standard brain template (MNI152) using an affine linear transformation with 12 DOF [90]. We resampled the aligned brains by a 4 mm kernel that generated 45 × 54 × 45 brain slices per time course. The rs-fMRI preprocessing pipeline produced 4-dimensional (4D) data, including time series within T∈[124,200] with the mode of 140 data points per participant; therefore, we obtained 4D NIfTI/NII files of 45 × 54 × 45 × T.

#### 3.3.2. Structural MRI

We preprocessed the structural MRI data from scratch using a 6-step pipeline where we first converted the DICOM raw images to NifTi/NII format using dcm2niix software [87]. Next, we extracted the brain regions by removing the skull and neck tissues from the data [88]. Then, using the FSL-VBM library [91], we segmented the brain images into grey matter (GM), white matter (WM), and cerebrospinal fluid (CSF). We used the GM images to register to the GM ICBM-152 standard template using a linear affine transformation with 6 DOF. Next, we concatenated the brain images, flipped them along the x-axis, then re-averaged to create a first-pass, study-specific template as a standard approach [88]. Next, we re-registered the structural MRI brains to the template using a non-linear transformation, and then resampled to create a 2 × 2 × 2 mm${}^{3}$ GM template in the standard space. Per FSL-VBM standard protocol, we applied a modulation technique to the structural MRI data by multiplying each voxel by the Jacobian of the warp field to compensate for the enlargement that occurred via the non-linear component of transformation. Subsequently, we used all the concatenated and averaged 3D GM images (one 3D sample per participant) to create a 4D data stack. Finally, we smoothed the structural MRI data using a range of Gaussian kernels with sigma = 3, 4 (FWHM of 4.6, 7, and 9.3 mm), as the research showed that the smoothing significantly impacted the performance of modeling [92,93]. The structural MRI preprocessing pipeline produced two sets (one set per sigma) of 3D NIfTI/NII files of 91 × 109 × 91.

### 3.4. Proposed Architecture: Optimized Vision Transformer (OViTAD)

Inspired by a transformer built for natural language processing use cases [94], vision transformers have been adopted for computer vision tasks such as image classification or object detection. The vanilla vision transformer [95] employs a dozen multi-head self-attention (MHSA) layers, considered the transformer blocks building the core of architecture. This algorithm splits an input image into small patches that are passed through positional embedding and transformer encoder layers. During the training process, the positional information is incorporated by attention layers which a similar to better predict farther data points from the current state [94,95]. The vision transformer generated patches from a given set of preprocessed images, converted the 2D arrays into 1D arrays, and decomposed them along the axes for the three channels. The dimension of each patch is calculated by multiplying the number of channels by the height and width of the patches. We prepared the linearly embedded arrays to feed into the next blocks. To address the objective of our multiple-class Alzheimer’s prediction, where we used specific imaging data dimensions; we set our transformer’s input dimension to 56 × 56 for fMRI and 112 × 112 for structural MRI, which were the closest meaningful dimensions reflecting popular image size. This data-driven approach allowed us to bypass a computationally massive grid search by optimizing the network’s hyperparameters. Since we reduced the vision transformer input dimension from 224 × 224 × 3 to 112 × 112 × 3 and 56 × 56 × 3, we reduced the number of heads in MHSA in architecture to optimize the architecture. The core intention was to improve the efficiency of our model while producing the same or better performance compared to the vanilla version with reduced trainable parameters. In the next step, the vision transformer used a positional embedding to feed the arrays to the transformer 8-head self-attention block with six layers in depth, which applied a set of standard steps to the arrays similar to the original architecture [94,95]. To decrease the chance of overfitting, we set our dropout and embedding dropout to 0.1. We used a multi-perceptron layer, known as the fully connected layer of 2048 neurons, to translate the features extracted by the optimal vision transformer to a format usable for the cross-entropy loss function to evaluate classification performance. Figure 1 pictures the architecture of the optimized vision transformer implemented in this study.

We used DeepViT, which is a deeper version of a vision transformer, to build our baselines [96]. DeepViT employs a mechanism known as re-attention, instead of MHSA, to reproduce attention maps to increase the diversity of features extracted by the architecture. The re-attention layers benefit from the interaction across various heads to capture further information, which improves the diversity of attention maps through a learnable transformation matrix known as Q. Figure 2 (left) demonstrates the DeepViT transformer block with its re-attention mechanism. To enhance the scope of our benchmarking, we used another vision transformer image classifier known as class-attention in image transformers (CaIT) that introduced a class-attention layer [97]. The CaIT architecture consists of two major components: (a) standard self-attention step which is identical to the ViT transformer, and (b) a class-attention layer step, including a set of operations to convert the positionally embedded patches into class embedding arrays (CLS), followed by a linear classification method. CaIT with the CLS mechanism avoids the saturation of deep vision transformers in the early state and allows the model to further learn across training. Figure 2 (right) shows the CaIT transform block.

### 3.5. fMRI Pipeline

We categorized the preprocessed 4D fMRI samples (one NIfTI/NII per participant) into AD, HC, and MCI classes. In the next step, we used a stratified split of 80%–10%–10% and randomly shuffled data at class level to generate three training, validation, and testing sets. Therefore, the sets included 226, 27, and 31 participants for training, validation, and testing. The main objective of this study was to perform a multiclass prediction; however, we expanded our modeling approach to explore the impact of merging MCI data with two other classes and generated samples for AD + MCI vs. HC and AD vs. HC + MCI experiments. Using our optimized model, we also built AD vs. HC and HC vs. MCI models for a consistent comparison with the literature. To perform a consistent comparison, we used the identical data splits generated for multiclassification for the two binary classifications, where we only modified the corresponding ground truth according to experiments.

#### 3.5.1. Data Decomposition from 4D to 2D

We decomposed the 4D fMRI data and z and t axes into 2D images using a lossless data conversion method to generate portable network graphics (PNG) samples for model development. We first loaded the NIfTI files into memory using the Nibabel package available at https://nipy.org/nibabel/, 20 August 2021 and employed the Python OpenCV library available at OpenCV.org to store the decomposed 2D images in the server. Next, we removed the last ten brain slices and empty brain images to improve data quality. To find the empty slices, we measured the sum of pixel intensity in a given brain image and only stored images with non-zero-sum. Equation (Equation 1) shows the details of fMRI data decomposition.
(1)$$\begin{array}{cc} \; & for\;\forall z=1\;to\;Z-10\\ \; & for\;\forall t=1\;to\;T\\ \; & if\;SI_{z,t}\left(BS_{z,t}\left(x,y\right)\right)=\sum_{X}^{x=1}\sum_{Y}^{y=1}BS\left(x,y\right)\neq 0:\\ \; & BS_{z,t}\left(x,y\right)\to PNG\left(BS_{z,t}\left(x,y\right)\right)\\ \; & otherwise:\\ \; & Ignore\;BS_{z,t}\left(x,y\right) \end{array}$$
where X, Y, and Z represent the spatial dimensions of fMRI data (45, 54, 45), and T refers to 140 data points of a given fMRI time course. SI(z,t) represents the sum of voxel intensity in a given brain slice, $BS_{z,t}\left(x,y\right)$ represents and PNG denotes the lossless data conversion function. The decomposition module produced 1,433,880 images consisting of 1,141,280, 138,600, and 154,000 samples for training, validation, and testing purposes.

#### 3.5.2. Modeling

The central objective of our fMRI pipeline was to address the multiclass prediction of AD, HC, and MCI using our designed optimal vision transformer. Furthermore, we considered two additional binary classification experiments mentioned earlier: (a) AD + MCI against HC and (b) HC + MCI against AD, to explore the clinical impact of merging MCI with the other classes. We built our optimized vision transformer (OViTAD) and three other baselines—CaIT, DeepViT, and vanilla vision transformer—and used the Amazon Web Services’ (AWS) SageMaker infrastructure as our development environment. We spun up a p3.8xlarge instance with 32 virtual central processing units (vCPUs) and 244 gigabyte (GB) memory. The instance included four NVIDIA TESLA V100-SXM2-16GB graphical processing units (GPUs) and 10 GB per second (Gbps) network performance. We trained all the models for 40 epochs and a batch size 64 using the Adam optimization method with a learning rate lr = 3 × 10^−5^, gamma = 0.7, and stepsize = 1. We monitored modeling performance across the epochs using accuracy rates and loss scores for training and validation sets. We used the accuracy rate of validation sets as the criteria for selecting the best model. We implemented the prediction module to load the stored best models into the memory and predict validation and testing sets with their probability scores at slice level. We evaluated the performance of the models using a standard classification report by calculating precision, recall, F1-score, and accuracy rates. Table A1 in the Appendix A demonstrates the models’ performance at slice level for validation and test datasets and three repetitions (random data splits) of fMRI experiments.

#### 3.5.3. Subject-Level Evaluation

We designed our modeling based on the decomposition of brain image into a 2D image; therefore, the performance obtained from the prediction module demonstrated the slice-level performance. To calculate the performance of our models at the subject level (see Table A1 Appendix A), we applied a vote for majority method to the predicted labels by aggregating the results based on subjects’ identifiers (IDs). Next, we calculated the probability of each class per subject and then voted for the class with the highest probability. Finally, we used our standard classification report to measure the performance of our models at the subject level. Table A2 in the Appendix A shows the models’ performance for validation and test datasets and three repetitions (random data splits) of experiments for fMRI data. First, we calculated the macro average (macro-ave) and weighted average (weighted-avg) for precision, recall, and F1-score evaluation metrics. Next, we analyzed at model level to explore classification performance across the experiments. We used the weighted average scores of the aforementioned four metrics and calculated each experiment’s average and standard deviation against three repetitions (random data splits). Table 2 shows the performance of models for validation and test sets with the averaged metrics and the corresponding standard deviation values. We summarized the results of this table in Figure A5 comparing the performance of fMRI models using averaged F1-score for three testing sets.

### 3.6. Structural Pipeline

#### 3.6.1. Data Split

We categorized the preprocessed 3D structural MRI samples (one NIfTI/NII per participant) into AD, HC, and MCI classes. In the next step, we used a stratified split of 80%–10%–10% and randomly shuffled data at class-level to generate three training, validation, and testing sets for two sets of preprocessed data S3 (sigma = 3 mm) and S4 (sigma = 4 mm). Therefore, the sets included 1167, 144, and 149 participants for training, validation, and testing, respectively. Similar to the fMRI pipeline, we explored the impact of merging MCI data with AD and HC. We used the identical data splits generated for multiclass prediction to address the binary classification experiments by updating the corresponding ground truth; this strategy allowed us to perform a consistent comparison across experiments and two sigma variations.

#### 3.6.2. Data Decomposition 3D to 2D

We employed the same technique explained in Equation Equation 1 to decompose 3D MRI data into 2D PNG images. As the structural MRI data are constructed without temporal information, we set the time parameter in the equation to T = 1. The structural MRI decomposition module produced 111,899 images per set containing 89,446, 11,040, and 11,413 samples for training, validating, and testing our models.

#### 3.6.3. Modeling

The main objective of the structural MRI pipeline was to conduct a multiclass prediction of AD, HC, and MCI classes using two sets of preprocessed data (sigma = 3, 4) and to evaluate our proposed optimal vision transformer architecture. Also, we used four other models as baselines similar to the fMRI pipeline to investigate the performance of optimal architecture. Furthermore, we considered combining MCI data with AD and HC to classify (a) AD + MCI against HC and (b) HC + MCI against AD. Similar to the fMRI pipeline, we utilized AWS SageMaker as the development environment on a p3.8xlarge instance equipped with NVIDIA GPUs. We trained all the models for 40 epochs and a batch size 64 using the Adam optimization method with a learning rate lr = 3 × 10^−5^, gamma = 0.7, and step_size = 1. Using loss scores and accuracy rates of training and validation sets, we evaluated the training process and selected the best model based upon the highest accuracy rate obtained from the validation sets. Since we designed our vision transformers to use 2D images, we developed a prediction module to output validation and test sets’ labels at slice level. We employed our standard classification report module to generate a macro and weighted average of precision, recall, F1-scores, and accuracy rates. We show the slice-level performance of structural MRI models in Table A3 and Table A4 in Appendix A and for sigma = 3, 4.

#### 3.6.4. Subject-Level

We used the predicted labels for brain slices and aggregated the results by the subject IDs to calculate the models’ performance at subject level; the slice-level performance is shown in Table A4
Appendix A. Then, using the postprocessing module based on the voting for majority concept, we counted the number of each class prediction in an experiment and measured each class probability. In the next step, we assigned the corresponding label of the highest probability to a given subject. Finally, we employed our standard classification reports as described earlier, and generated the evaluation scores at the subject level. Table A5 and Table A6 in the Appendix A demonstrate the subject-level performance of structural MRI models for preprocessed data with spatial smoothing sigma = 3, 4, respectively. To measure the performance of experiments at the model level, we used the weighted average evaluation scores and calculated the average and standard deviation of the scores for both structural MRI datasets, shown in Table 3. We summarized the results of this table in Figure A6 comparing the performance of sMRI models using averaged F1-score for three testing sets.

### 3.7. Discussion

#### 3.7.1. Technical/Architecture Design

We designed an optimized vision transformer architecture to predict multiple stages of Alzheimer’s disease using fMRI and MRI data. Our end-to-end pipeline for two modalities was built on four major components: (a) aggressive preprocessing of fMRI and MRI data, (b) data decomposition from higher dimensions to 2D, (c) vision transformer model development, and (d) postprocessing. The core concept of this study was to explore the capability of vision transformers to predict Alzheimer’s stages. We exhaustively trained models to conduct a comprehensive evaluation of our proposed architecture. We investigated the performance of our baselines and our proposed architecture against fMRI and two sets of structural MRI data to address the 3-class AD vs. HC vs. MCI, AD vs. HC + MCI, and AD + MCI vs. HC classifications. To demonstrate the robustness of our modeling approach, we repeated each experiment with random data splits three times. More random data splits, such as five to ten runs, could be explored in future work. We reported the performances at the slice level and subject level, which led us to compare our models across all experiments (model level). We proposed an optimized vision transformer architecture as the core of our end-to-end prediction pipeline. Our optimization approach is based on the scientific fact of using an image input size of architecture that has the closest and most meaningful input dimensions of preprocessed fMRI data. Therefore, we set the architecture input dimension to 56 × 56 and resample our data (45 × 54) to fit our optimal architecture, where the originality of data content remains through minimal upsampling. Next, we consider reducing the number of heads in the multi-head attention layer to decrease the complexity and trainable parameters of the network. We showed in Table 4 that we decreased the input image size and trainable parameters in the optimized network by 75% and 28% compared to the vanilla vision transformer, while improving the models’ performance in the fMRI experiments and producing a similar performance to other models in the structural MRI experiments. Unlike grid search-based optimization, which requires massive model development to achieve an optimal architecture and topology, our fact- and data-driven optimization method, which stems from the impact of input size, produced faster converging modeling. This allowed us to explore a broader set of model development and clinical analysis.

We consider the fMRI testing datasets as our gold standard to compare the performance of our models. Unlike training and validation datasets, the testing datasets are unseen and never used in the training processes. The models’ performance at the subject. level using fMRI data, shown in Table 3, reveals that OViTAD, DeepViT, ViT-vanilla, and ViT-224-8 in AD-HCMCI classification outperforms other models with an F1-score of 0.99±0.02. Also, among the models trained for the 3-class AD vs. HC vs. MCI prediction, our optimized OViTAD model is on par with the ViT-vanilla, and ViT-224-8 outperforms other models with an F1-score of 0.97±0.02; our optimized models contain much fewer trainable parameters than other models. Also, we investigated the impact of the postprocessing step developed based on voting for the majority algorithm, and the results indicated that the models’ performance at the subject level (after postprocessing) with an averaged F1-score of 0.89±0.02 across all experiments (testing datasets) are higher (with 3% improvement) than slice-level ones with an averaged F1-score of 0.86±0.02. This finding aligned with the literature [54,58,64] that shows that postprocessing plays a crucial role in improving the performance of modeling and proves that decomposition of data from higher dimensions to 2D and back-transforming the slice-level predictions to the subject level improve the quality of prediction significantly.

Similar to the above approach, we consider the structural MRI (sigma = 3) testing datasets as the golden standard to investigate the best-performing model. The results shown in Table 3 reveal that our OViTAD is on par with DeepViT, ViT-vanilla, and ViT-224-8 in the ADMCI−HC S3 and S4 experiments in terms of F1-scores at the subject level. To explore the central objective of this study, we reviewed the performance of models for 3-class AD vs. HC vs. MCI prediction. The results indicated that ViT-vanilla and ViT-224-8 competed with our OViTAD and produced an F1-score of 0.99±0.01 (0.004 is negligibly higher than OViTAD) using MRI S3. After preprocessing, the original MRI dimension was 91X109, and we downsampled the structural MRI data to 112 × 112, causing a loss in contextual information. Similarly, we analyzed the behavior of our models trained and evaluated by the preprocessed MRI with sigma = 4 testing datasets. Our OViTAD model using MRI S4 was on par with other architectures, producing the best performance with an F1-score of 0.99±0.01.

The results suggest that the input size and number of patches in the attention layers greatly impact the performance of the structural MRI models. In a similar observation to fMRI testing datasets, the models’ performance at the subject level (after postprocessing with voting for a majority) increased by 7% compared to the slice-level models across the experiments for sigma = 3, 4. Our analysis indicated that spatial smoothing with a Gaussian kernel of sigma = 3 mm resulted in slightly higher evaluation scores across the study (an average increase of 0.43% in sigma = 3 compared to the sigma = 4 dataset) which aligns with the previous research; however, the improvement is negligible [54,58]. Spatial smoothing is important in preprocessing MRI data that removes random noise in a given voxel’s neighborhoods [98,99].

This finding implies that the nature of features extracted by attention layers in the vision transformer should differ from the features extracted by convolutional layers since the impact of sigma = 3, 4 in the previous studies was negligible [54,58]. Next, we calculated the confusion matrix of testing samples normalized per group for the best-performing OViTAD fMRI (test set 2), MRI-S3, and MRI-S4 (test sets 3) models in the multiple classification experiment to predict AD vs. HC vs. MCI, illustrated in Figure 3. The performance of the best-performing OViTAD models for the same test sets across 40 epochs is shown in Figure A4, Appendix A.

Moreover, we comprehensively compared our findings and the recent literature reviews in which the ADNI dataset was used for Alzheimer’s disease classification. We carefully selected the most current, highly referenced studies and offered novel techniques where the performance of models was highly competitive. Table 5 compares the performance achieved by OViTAD in the two modalities with the most highly referenced recent literature. Our finding shows that this study offers a broader range of classifications where the optimized vision transformer outperforms the state-of-the-art models.

#### 3.7.2. Clinical Observation

We considered combining the health control brains with Alzheimer’s and mild cognitive impairment brains to generate new sets from the ADNI dataset to perform two binary classification tasks using all the models. The fMRI models revealed a consistent pattern in which the AD vs. HC + MCI models outperformed AD + MCI vs. HC by 4.64% with respect to averaged F1-scores across all experiments shown in Table 2. This finding revealed some level of similarity between HC and MCI functional data. Also, the results showed that our predictive models could differentiate HC data from non-HC data, which revealed that our models properly addressed the aging effect in this study. Furthermore, we analyzed the binary models trained by structural MRI data for AD + MCI vs. HC and AD vs. HC + MCI experiments for the two sigma = 3,4. The results indicated that our HC vs. AD + MCI models outperformed AD vs. HC + MCI by 2.82%, respecting the averaged F1-scores across all experiments for the two sigma values shown in Table 3.

#### 3.7.3. Local and Global Attention Visualization

We extracted the attention weights and produced post-SoftMax for eight self-attention heads with a depth of six. Then, using a random AD fMRI brain slice, we generated the self-attention maps based on OViTAD for AD vs. HC vs. MCI classification as shown in Figure A1, Appendix A. The attention maps in each column represent one self-attention head, whereas the maps in each row represent the depth of attention layers. Also, we explored the impact of attention mechanisms at the global level. We utilized the last feature vector of OViTAD—the fMRI AD vs. HC vs. MCI classification, which is a fully connected layer (FC)—and considered it the global attention feature. The FC layer represents the features produced by the self-attention layers; therefore, it contains the information of global attention. We employed an element-wise operator to obtain the sum of multiplication between each pixel and all the elements in the FC vector. Next, we generated the normalized global attention feature maps for a set of AD fMRI slices in the testing set as shown in Equation (Equation 2) and visualized the maps using the CIVIDIS color map, illustrated in Figure 4.
(2)$$\begin{array}{cc} \; & image_{resize}=Resize\left(image_{original}\to 56\times 56\right)\\ \; & GlobalAttentionFeatureMap\left(GAFM\right)=\sum image_{resize}\times FC_{vector}\\ \; & GAFM_{normalized}=\left(GAFM-min\left(GAFM\right)\right)*\frac{255}{max(GAFM)-min(GAFM)} \end{array}$$

#### 3.7.4. Limitations

The number of repetitions for model development is considered a limitation in this research study. Although we utilized a large dataset and generated three random data splits for modeling, it is highly recommended to repeat this exercise up to 10 times with randomly shuffled data to ensure the robustness of OViTAD. Also, we included voting for a majority technique as postprocessing to stabilize models’ performance; however, this step would add an extra layer of computation to our pipeline, increasing the modeling cost. Future work could address such a limitation using upper-dimension models, including 3D vision transformers. Training of vision transformer models is costly; therefore, reducing the image input size discussed in this research decreases training time and inspection latency. Finally, this research study outlined an end-to-end machine learning pipeline to predict Alzheimer’s disease stages using the ADNI dataset so that the models’ performance reflects the accuracy of the pipeline for this dataset. Since the early prediction of this brain disorder is crucial in clinical studies, a variety of existing Alzheimer’s datasets should be explored along with ADNI to examine OViTAD performance in future work.

## 4. Conclusions

This study introduced an optimized vision transformer called OViTAD to predict healthy, MCI, and AD brains using rs-fMRI and structural MRI (sigma = 3,4 mm) data. The prediction pipeline included two separate preprocessing stages for the two modalities, training and evaluation of slice-level vision transformers and a postprocessing step based on voting for the majority concept. The results showed that our optimized vision transformer outperformed and was on par with the vision transformers-based benchmark. OViTAD 30% reduced the number of trainable parameters compared to the vanilla ViT. The average performance of OViTAD across three repetitions (random data splits) was 97% ± 0.0 and 99.55% ± 0.39 for the two modalities for the multi-class classification experiments, which outperformed most existing deep learning and CNN-based models. Also, we introduced a method of visualizing the attention mechanism’s global effect, enabling scientists to explore crucial brain areas. This study showed that the vision transformers could outperform and compete with the state-of-the-art algorithms to predict various stages of Alzheimer’s disease with less complex architectures.

## Figures and Tables

**Figure 1 brainsci-13-00260-f001:**
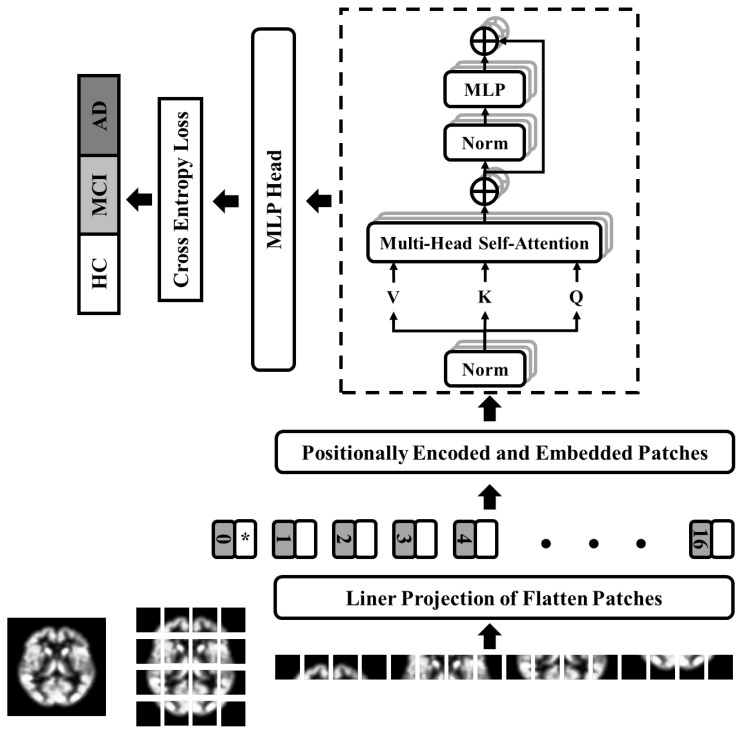
The OViTAD architecture is an optimized ViT shown for structural MRI data composed of a linear projection layer applied to the flattened patches fed into an 8-HSA Transformer. The MLP layer of 2048 parameters translates the features from the transformer encoder to a proper format for the cross-entropy loss function.

**Figure 2 brainsci-13-00260-f002:**
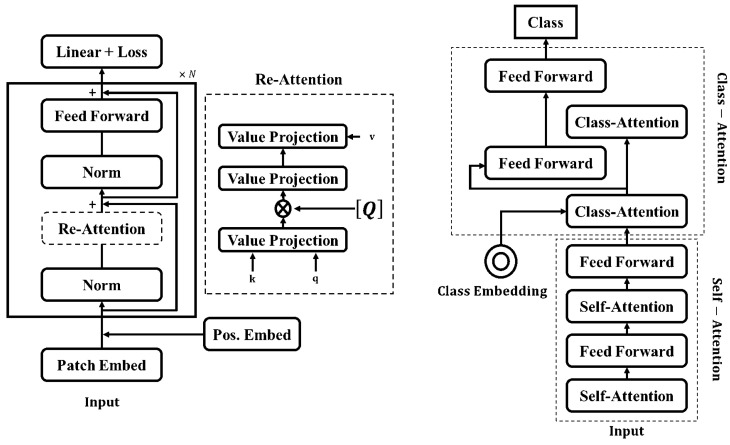
The transformer block in DeepViT architecture includes a re-attention module instead of a standard self-attention layer (**Left**). Class-Attention in Image Transformer architecture consists of a class embedding (CLS) and additional class-attention layers preceded by self-attention layers (**Right**).

**Figure 3 brainsci-13-00260-f003:**
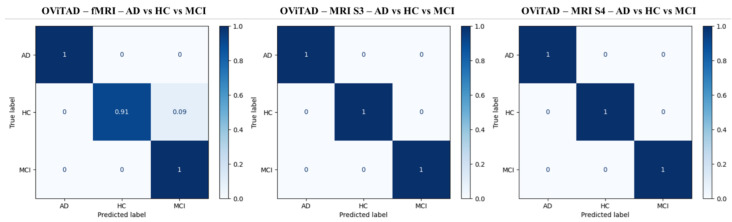
The normalized confusion matrices for the best-performing fMRI (**left**), MRI-S3 (**middle**), and MRI-S4 (**right**) OViTAD models to classify AD vs. HC vs. MCI at subject-level.

**Figure 4 brainsci-13-00260-f004:**
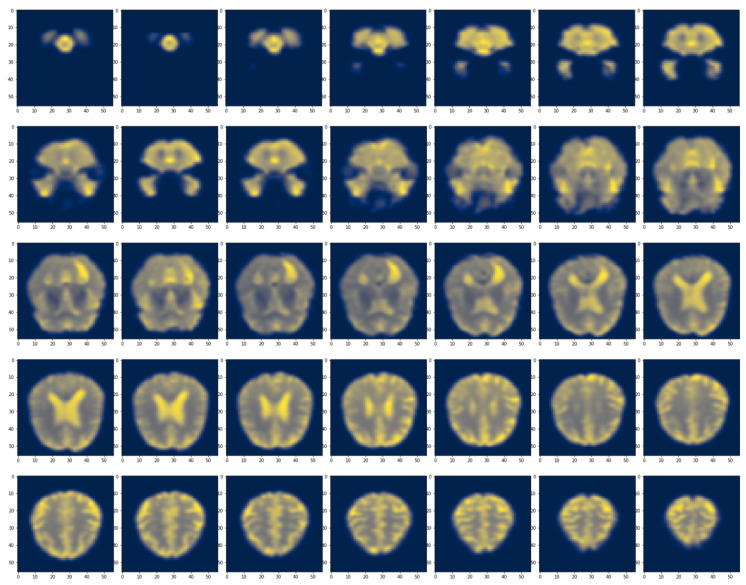
The global attention feature map was obtained by multiplying the FC layer vector by each pixel in the fMRI brain slices and measuring the sum of the multiplication per pixel. Next, we normalized the feature map to (0, 255) and visualized the maps using the CIVIDIS color map. Finally, we selected the first slice of each time-course to demonstrate various brain morphology across the fMRI data acquisition.

**Table 1 brainsci-13-00260-t001:** The demographic of two sets of ADNI data used in model development shows all the groups are older adults within an age group of >75.

Modality	Total	Group	Participant	Female	Age	Male	Age	MMSE
rs-fMRI	284	AD	54	27	80.96 ± 4.64	27	79.0 ± 2.74	22.70 ± 2.10
		HC	99	49	79.78 ± 4.76	50	82.57 ± 3.88	28.82 ± 1.35
		MCI	131	66	79.15 ± 3.09	65	79.72 ± 4.84	26.53 ± 2.51
MRI	1460	AD	577	232	80.98 ± 4.65	345	81.27 ± 4.08	23.07 ± 2.06
		HC	108	51	79.37 ± 3.54	57	80.81 ± 4.42	28.81 ± 1.35
		MCI	775	265	80.28 ± 3.31	510	81.61 ± 4.15	26.53 ± 2.09

**Table 2 brainsci-13-00260-t002:** To evaluate the performance of experiments referring to model-level results, we used the weighted average scores of subject-level results and calculated each experiment’s average and standard deviation across three repetitions for validation and test sets (random data splits).

Model	Dataset	Precision	Recall	F1-Score	Accuracy
CaIT_ADMCI-HC	Val	0.53 ± 0.14	0.68 ± 0.02	0.57 ± 0.07	0.68 ± 0.02
	Test	0.54 ± 0.21	0.66 ± 0.02	0.53 ± 0.04	0.66 ± 0.02
CaIT_AD-HCMCI	Val	0.66 ± 0	0.81 ± 0	0.73 ± 0	0.81 ± 0
	Test	0.65 ± 0	0.81 ± 0	0.72 ± 0	0.81 ± 0
CaIT_AD-HC-MCI	Val	0.4 ± 0.14	0.54 ± 0.06	0.43 ± 0.11	0.54 ± 0.06
	Test	0.37 ± 0.01	0.46 ± 0.02	0.37 ± 0.01	0.46 ± 0.02
DeepViT_AD_HC_MCI	Val	1 ± 0	1 ± 0	1 ± 0	1 ± 0
	Test	0.96 ± 0.02	0.96 ± 0.02	0.96 ± 0.02	0.96 ± 0.02
DeepViT_AD_HCMCI	Val	0.99 ± 0.02	0.99 ± 0.02	0.99 ± 0.02	0.99 ± 0.02
	Test	0.99 ± 0.02	0.99 ± 0.02	0.99 ± 0.02	0.99 ± 0.02
DeepViT_ADMCI_HC	Val	1 ± 0	1 ± 0	1 ± 0	1 ± 0
	Test	0.99 ± 0.02	0.99 ± 0.02	0.99 ± 0.02	0.99 ± 0.02
ViT_224_8_AD_HC_MCI	Val	0.99 ± 0.02	0.99 ± 0.02	0.99 ± 0.02	0.99 ± 0.02
	Test	0.97 ± 0.03	0.97 ± 0.03	0.97 ± 0.03	0.97 ± 0.03
ViT_224_8_AD_HCMCI	Val	1 ± 0	1 ± 0	1 ± 0	1 ± 0
	Test	0.99 ± 0.02	0.99 ± 0.02	0.99 ± 0.02	0.99 ± 0.02
ViT_224_8_ADMCI_HC	Val	1 ± 0	1 ± 0	1 ± 0	1 ± 0
	Test	0.97 ± 0	0.97 ± 0	0.97 ± 0	0.97 ± 0
ViT_vanilla_AD_HC_MCI	Val	0.99 ± 0.02	0.99 ± 0.02	0.99 ± 0.02	0.99 ± 0.02
	Test	0.97 ± 0	0.97 ± 0	0.97 ± 0	0.97 ± 0
ViT_vanilla_AD_HCMCI	Val	1 ± 0	1 ± 0	1 ± 0	1 ± 0
	Test	0.99 ± 0.02	0.99 ± 0.02	0.99 ± 0.02	0.99 ± 0.02
ViT_vanilla_ADMCI_HC	Val	1 ± 0	1 ± 0	1 ± 0	1 ± 0
	Test	0.98 ± 0.02	0.98 ± 0.02	0.98 ± 0.02	0.98 ± 0.02
OViTAD_AD_HC_MCI	Val	0.99 ± 0.02	0.99 ± 0.02	0.99 ± 0.02	0.99 ± 0.02
	Test	0.97 ± 0	0.97 ± 0	0.97 ± 0	0.97 ± 0
OViTAD_AD_HCMCI	Val	0.99 ± 0.02	0.99 ± 0.02	0.99 ± 0.02	0.99 ± 0.02
	Test	0.99 ± 0.02	0.99 ± 0.02	0.99 ± 0.02	0.99 ± 0.02
OViTAD_ADMCI_HC	Val	1 ± 0	1 ± 0	1 ± 0	1 ± 0
	Test	0.98 ± 0.02	0.98 ± 0.02	0.98 ± 0.02	0.98 ± 0.02
OViTAD_AD_HC	Val	1 ± 0	1 ± 0	1 ± 0	1 ± 0
	Test	0.99 ± 0.02	0.99 ± 0.02	0.99 ± 0.02	0.99 ± 0.02
OViTAD_HC_MCI	Val	1 ± 0	1 ± 0	1 ± 0	1 ± 0
	Test	0.97 ± 0.03	0.97 ± 0.03	0.97 ± 0.03	0.97 ± 0.03

**Table 3 brainsci-13-00260-t003:** The models’ performance of two sets for structural MRI experiments evaluated by standard evaluation metrics.

Model	Dataset	Precision	Recall	F1-Score	Accuracy
CaIT_S3_AD-HC-MCI	Val	0.74 ± 0.02	0.8 ± 0.02	0.77 ± 0.02	0.8 ± 0.02
	Test	0.71 ± 0.02	0.77 ± 0.03	0.73 ± 0.03	0.77 ± 0.03
CaIT_S3_AD-HCMCI	Val	0.72 ± 0.02	0.72 ± 0.03	0.7 ± 0.03	0.72 ± 0.03
	Test	0.71 ± 0.02	0.7 ± 0.01	0.69 ± 0.01	0.7 ± 0.01
CaIT_S3_ADMCI-HC	Val	0.87 ± 0	0.93 ± 0	0.9 ± 0	0.93 ± 0
	Test	0.85 ± 0	0.92 ± 0	0.88 ± 0	0.92 ± 0
DeepViT_S3_AD-HC-MCI	Val	0.81 ± 0.06	0.85 ± 0.03	0.82 ± 0.04	0.85 ± 0.03
	Test	0.77 ± 0.02	0.84 ± 0.02	0.81 ± 0.02	0.84 ± 0.02
DeepViT_S3_AD-HCMCI	Val	0.84 ± 0.04	0.84 ± 0.05	0.84 ± 0.05	0.84 ± 0.05
	Test	0.83 ± 0.04	0.83 ± 0.04	0.83 ± 0.04	0.83 ± 0.04
DeepViT_S3_ADMCI-HC	Val	0.87 ± 0	0.93 ± 0	0.9 ± 0	0.93 ± 0
	Test	0.85 ± 0	0.92 ± 0	0.88 ± 0	0.92 ± 0
ResNet50_S3_AD-HC-MCI	Val	0.85 ± 0.09	0.86 ± 0.05	0.84 ± 0.06	0.86 ± 0.05
	Test	0.83 ± 0.06	0.85 ± 0.02	0.82 ± 0.02	0.85 ± 0.02
ResNet50_S3_AD-HCMCI	Val	0.84 ± 0.01	0.84 ± 0.01	0.84 ± 0.01	0.84 ± 0.01
	Test	0.84 ± 0.04	0.84 ± 0.04	0.84 ± 0.04	0.84 ± 0.04
ResNet50_S3_ADMCI-HC	Val	0.87 ± 0	0.93 ± 0	0.9 ± 0	0.93 ± 0
	Test	0.85 ± 0	0.92 ± 0	0.88 ± 0	0.92 ± 0
ViT_S3_AD-HC-MCI	Val	0.84 ± 0.05	0.88 ± 0.02	0.85 ± 0.03	0.88 ± 0.02
	Test	0.84 ± 0.08	0.85 ± 0.03	0.83 ± 0.04	0.85 ± 0.03
ViT_S3_AD-HCMCI	Val	0.84 ± 0.01	0.84 ± 0.01	0.83 ± 0.02	0.84 ± 0.01
	Test	0.84 ± 0.03	0.83 ± 0.02	0.83 ± 0.02	0.83 ± 0.02
ViT_S3_ADMCI-HC	Val	0.87 ± 0	0.93 ± 0	0.9 ± 0	0.93 ± 0
	Test	0.85 ± 0	0.92 ± 0	0.88 ± 0	0.92 ± 0
OViTAD_S3_AD-HC-MCI	Val	0.78 ± 0.02	0.84 ± 0.02	0.81 ± 0.02	0.84 ± 0.02
	Test	0.75 ± 0.03	0.82 ± 0.03	0.79 ± 0.03	0.82 ± 0.03
OViTAD_S3_AD-HCMCI	Val	0.79 ± 0.03	0.77 ± 0.05	0.75 ± 0.07	0.77 ± 0.05
	Test	0.79 ± 0.02	0.77 ± 0.04	0.75 ± 0.06	0.77 ± 0.04
OViTAD_S3_ADMCI-HC	Val	0.87 ± 0	0.93 ± 0	0.9 ± 0	0.93 ± 0
	Test	0.85 ± 0	0.92 ± 0	0.88 ± 0	0.92 ± 0
OViTAD_S3_AD-HC	Val	1 ± 0	1 ± 0	1 ± 0	1 ± 0
	Test	1 ± 0	1 ± 0	1 ± 0	1 ± 0
OViTAD_S3_HC-MCI	Val	1 ± 0	1 ± 0	1 ± 0	1 ± 0
	Test	1 ± 0	1 ± 0	1 ± 0	1 ± 0
CaIT_S4_AD-HC-MCI	Val	0.84 ± 0.03	0.9 ± 0.03	0.87 ± 0.03	0.9 ± 0.03
	Test	0.81 ± 0.01	0.88 ± 0.01	0.84 ± 0.01	0.88 ± 0.01
CaIT_S4_AD-HCMCI	Val	0.87 ± 0.02	0.87 ± 0.02	0.87 ± 0.02	0.87 ± 0.02
	Test	0.86 ± 0.02	0.86 ± 0.02	0.86 ± 0.02	0.86 ± 0.02
CaIT_S4_ADMCI-HC	Val	0.87 ± 0	0.93 ± 0	0.9 ± 0	0.93 ± 0
	Test	0.85 ± 0	0.92 ± 0	0.88 ± 0	0.92 ± 0
DeepViT_S4_AD-HC-MCI	Val	0.85 ± 0.01	0.91 ± 0.01	0.88 ± 0.01	0.91 ± 0.01
	Test	0.82 ± 0.01	0.88 ± 0.01	0.85 ± 0.01	0.88 ± 0.01
DeepViT_S4_AD-HCMCI	Val	0.91 ± 0.01	0.91 ± 0.01	0.91 ± 0.01	0.91 ± 0.01
	Test	0.89 ± 0.02	0.89 ± 0.02	0.89 ± 0.02	0.89 ± 0.02
DeepViT_S4_ADMCI-HC	Val	0.87 ± 0	0.93 ± 0	0.9 ± 0	0.93 ± 0
	Test	0.85 ± 0	0.92 ± 0	0.88 ± 0	0.92 ± 0
ResNet50_S4_AD-HC-MCI	Val	0.93 ± 0.01	0.93 ± 0.01	0.91 ± 0.01	0.93 ± 0.01
	Test	0.89 ± 0.06	0.91 ± 0.02	0.89 ± 0.03	0.91 ± 0.02
ResNet50_S4_AD-HCMCI	Val	0.93 ± 0	0.93 ± 0	0.93 ± 0	0.93 ± 0
	Test	0.91 ± 0.01	0.91 ± 0.01	0.91 ± 0.01	0.91 ± 0.01
ResNet50_S4_ADMCI-HC	Val	0.87 ± 0	0.93 ± 0	0.9 ± 0	0.93 ± 0
	Test	0.87 ± 0.05	0.92 ± 0	0.89 ± 0.01	0.92 ± 0
ViT_S4_AD-HC-MCI	Val	0.9 ± 0.06	0.91 ± 0.02	0.89 ± 0.02	0.91 ± 0.02
	Test	0.88 ± 0.06	0.9 ± 0.02	0.87 ± 0.02	0.9 ± 0.02
ViT_S4_AD-HCMCI	Val	0.91 ± 0	0.91 ± 0	0.91 ± 0	0.91 ± 0
	Test	0.9 ± 0.03	0.9 ± 0.03	0.9 ± 0.03	0.9 ± 0.03
ViT_S4_ADMCI-HC	Val	0.87 ± 0	0.93 ± 0	0.9 ± 0	0.93 ± 0
	Test	0.85 ± 0	0.92 ± 0	0.88 ± 0	0.92 ± 0
OViTAD_S4_AD-HC-MCI	Val	0.86 ± 0.06	0.9 ± 0.02	0.87 ± 0.03	0.9 ± 0.02
	Test	0.81 ± 0.01	0.87 ± 0.01	0.84 ± 0.01	0.87 ± 0.01
OViTAD_S4_AD-HCMCI	Val	0.9 ± 0	0.89 ± 0	0.89 ± 0	0.89 ± 0
	Test	0.89 ± 0.02	0.88 ± 0.02	0.88 ± 0.02	0.88 ± 0.02
OViTAD_S4_ADMCI-HC	Val	0.87 ± 0	0.93 ± 0	0.9 ± 0	0.93 ± 0
	Test	0.85 ± 0	0.92 ± 0	0.88 ± 0	0.92 ± 0
OViTAD_S4_AD-HC	Val	1 ± 0	1 ± 0	1 ± 0	1 ± 0
	Test	1 ± 0	1 ± 0	1 ± 0	1 ± 0
OViTAD_S4_HC-MCI	Val	1 ± 0	1 ± 0	1 ± 0	1 ± 0
	Test	1 ± 0	1 ± 0	1 ± 0	1 ± 0

**Table 4 brainsci-13-00260-t004:** The number of trainable parameters reduced by 28% compared to vanilla vision transformer and DeepViT while producing higher performance in fMRI and similar performance to other models in structural MRI data.

Model	Input (Channel,x,y)	Params
CaIT	3,224,224	120,707,075
DeepViT	3,224,224	53,532,867
ViT-vanilla	3,224,224	53,532,675
ViT-224-8	3,224,224	40,949,763
OViTAD	3,56,56	38,406,147

**Table 5 brainsci-13-00260-t005:** Comparison between recent studies of Alzheimer’s classification using ADNI and our OViTAD. The analysis shows that our study addresses a broader classification aspect with novel vision transformer technology, and our model performance outperformed the literature. Further details is found at Table A7.

Reference	Modality	AD vs. HC vs. MCI	AD + MCI vs. HC	AD vs. MCI+HC	AD vs. HC	MCI vs. HC
Lin et al. 2018 [100]	MRI	-	-	-	88.79%	-
Dimitriadis et al. 2018 [101]	MRI	61.90%	-	-	-	-
Kruthika et al. 2019 [102]	MRI	90.47%	-	-	-	-
Spasov et al. 2019 [103]	MRI + Clinical	-	-	-	-	86%
Basaia et al. 2019 [104]	MRI	-	-	-	98%	87%
Abrol et al. 2020 [105]	MRI	83.01%	-	-	-	-
Shao et al. 2020 [106]	MRI+PET	-	-	-	92.51%	82.53%
Alinsaif et al. 2021 [107]	MRI	-	70.50%	-	62.22%	-
Alinsaif et al. 2021 [107]	MRI	-	91.61%	-	92.78%	-
Ramzan et al. 2019 [63]	rs-fMRI	97.92%	-	-	-	-
Hojjati et al. 2018 [108]	MRI + rs-fMRI	-	-	93%	-	-
Cui et al. 2019 [109]	MRI	-	-	-	91.33%	-
Amoroso et al. 2018 [110]	MRI	38.80%	-	-	-	-
Buvaneswari et al. 2021 [111]	rs-fMRI	-	-	-	-	79.15%
Duc et al. 2019 [61]	rs-fMRI+Clinical	-	-	-	85.27%	-
OViTAD—fMRI	rs-fMRI	0.97 ± 0	0.98 ± 0.02	0.99 ± 0.02	0.99 ± 0.02	0.97 ± 0.03
OViTAD—MRI (Sigma = 3)	MRI	0.9955% ± 0.0039	1 ± 0	0.9955 ± 0.0039	1 ± 0	1 ± 0
OViTAD—MRI (Sigma = 4)	MRI	0.9955% ± 0.0039	1 ± 0	0.9955 ± 0.0039	1 ± 0	1 ± 0

## Data Availability

Not applicable.

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
