# Peer review of "OViTAD: Optimized Vision Transformer to Predict Various Stages of Alzheimer’s Disease Using Resting-State fMRI and Structural MRI Data"

_brainsci, 2023, doi:10.3390/brainsci13020260_

Round 1

Reviewer 1 Report

Thank you for inviting me to review this manuscript.
In this study the authors aimed to introduce an optimized vision transformer architecture to predict the group membership by separating healthy adults, mild cognitive impairment, and Alzheimer’s’ brains within the same age group from resting-state functional and structural magnetic resonance data. This looks a promising and exciting new approach for neuroimaging applications, especially for Alzheimer’s disease prediction.
This work is very useful and the method look to me very robust and rigorous, the paper is well written and sufficiently detailed.
I only have few suggestions for improving it:
1) The authors could add a figure summarizing Table 2 to give more prominence and visibility to the results.
2) The authors could emphasize more the importance of their findings in the conclusions and suggest future perspectives.

Author Response

Dear respected Reviewer 1, 

We thank you and the respected reviewers for your time and effort in reviewing our manuscript. The feedback has been invaluable in improving the content and presentation of our paper.

We have carefully reviewed the comments and revised our manuscript according to the reviewers' comments. The changes are highlighted in the attached manuscript, and our point-by-point responses are provided in the letter below. Also, our native English-speaking co-authors reviewed the manuscript and applied the necessary changes to improve the content of the manuscript.

On behalf of the team, I affirm that all authors have read and approved the changes made to the manuscript. We hope that the revised paper is now suitable for inclusion in MDPI Brain Sciences, and we look forward to hearing from you.

Yours sincerely,

Saman Sarraf, PhD, SMIEEE

Director, Machine Learning at Johnson & Johnson

Post-doctoral Fellowship at Stanford University

Ex-chair for Senior Member Committee at Silicon Valley Chapter of IEEE

Reviewer 2 Report

The article is about using optimized vision transformer to predict various stages of Alzheimer's disease. The authors did a great job in reporting obtained results and there is no doubt, they have the needed capability and other resources for the job.

Observations:

1. This work should be repackaged as a modification or extension of existing architectures/structures e.g. DeepVIT, CaIT as they were used as baseline developments (lines 231 and 239). Another reason is stated in Line 400 (equal results)
2. General readers may not know what a vision transformer is, it will enrich the article if the Vision Transformer is discussed with 2 lines and probably with a mathematical equation. Lines 211-212,

3. Line 137: ... suppress the effect of aging ... needs to be clarified to avoid misleading conclusion/understanding 

4. As with observation 1, the authors in line 162 acknowledged that enhanced multi level preprocessing is the boon of the work. Indicating this in the abstract and title will be very appropriate

5. Line 220 - 221:  ...from 224 to 56 and 112 ... , is not clear  

6. Line 261-261 ... sentence with multiple 'represented' ... needs correction/update

7. The point starting from line 267 was earlier made in Lines 253-4. The use of Furthermore in line 267 ... may need a redress, e.g, recall, or earlier mentioned

8. Components listed on Line 356 should include the preprocessing 

9. Line 364: (randomly data splits) --> (with randomly ...)??

10. Line 373: Table ??

11. The group sets on line 444 - 5 are not different 

Author Response

Dear respected reviewer 2, 

We thank you and the respected reviewers for your time and effort in reviewing our manuscript. The feedback has been invaluable in improving the content and presentation of our paper.

We have carefully reviewed the comments and revised our manuscript according to the reviewers' comments. The changes are highlighted in the attached manuscript, and our point-by-point responses are provided in the letter below. Also, our native English-speaking co-authors reviewed the manuscript and applied the necessary changes to improve the content of the manuscript.

On behalf of the team, I affirm that all authors have read and approved the changes made to the manuscript. We hope that the revised paper is now suitable for inclusion in MDPI Brain Sciences, and we look forward to hearing from you.

Yours sincerely,

Saman Sarraf, PhD, SMIEEE

Director, Machine Learning at Johnson & Johnson

Post-doctoral Fellowship at Stanford University

Ex-chair for Senior Member Committee at Silicon Valley Chapter of IEEE

Reviewer 3 Report

This manuscript presents an application of vision transformers on ADNi data for classifying s/fMRI data into healthy control, MCI, and AD. There are several major issues regarding the organization and presentations, technical soundness, novelty, and significance of this work:

- The text is in general very difficult to read. The authors should split the texts (especially in the introduction and related work sections) into paragraphs to facilitate fluency of the text for the readers. The text switches from present to past tense from time to time. The language of the text definitely can be improved. There are also many repetitive texts and redundancy, especially regarding the fMRI and sMRI pipelines.

- The experiments are repeated 3 times and the mean and standard deviation across 3 runs are reported. 3 runs of experiments are not enough to reliably estimate the standard deviation. At least 10 repetitions with random splits are needed.

- The novelty of the presented approach is very limited and is summarized as reducing the input size of the vision transformer (from 224*224 to 56*56 or 112*112). This is not enough contribution as this reduction did not even improve the performance compared to vanilla architecture.

- Some arbitrary choices have been made to engineer the data and make it compatible with input images to a vision transformer. For example, the authors opted to convert f/sMRI data into PNG format (!), or use 2d-slices instead of full 3D or 4D data. These choices limit the future application of the method.

- Given the presented results, the proposed method works more or less the same as other competitive methods. This puts the significance of the results under question.

- There are several non-standard arbitrary choices made for evaluating the results at the subject level (e.g., using majority voting for subject-level inference) that make the judgment about the validity of results very difficult. Add to this close to perfect classification performance in almost all cases which shows either the problem is easy to solve or unfair performance evaluation.

Author Response

Dear respected reviewer 3, 

We thank you and the respected reviewers for your time and effort in reviewing our manuscript. The feedback has been invaluable in improving the content and presentation of our paper.

We have carefully reviewed the comments and revised our manuscript according to the reviewers' comments. The changes are highlighted in the attached manuscript, and our point-by-point responses are provided in the letter below. Also, our native English-speaking co-authors reviewed the manuscript and applied the necessary changes to improve the content of the manuscript.

On behalf of the team, I affirm that all authors have read and approved the changes made to the manuscript. We hope that the revised paper is now suitable for inclusion in MDPI Brain Sciences, and we look forward to hearing from you.

Yours sincerely,

Saman Sarraf, PhD, SMIEEE

Director, Machine Learning at Johnson & Johnson

Post-doctoral Fellowship at Stanford University

Ex-chair for Senior Member Committee at Silicon Valley Chapter of IEEE

Round 2

Reviewer 2 Report

The authors addressed all my concerns in this update.

Thank you.

Author Response

Thanks very much for reviewing our manuscript.